# Electrochemical and X-ray Examinations of Erosion Products during Dressing of Superhard Grinding Wheels Using Alternating Current and Ecological Electrolytes of Low Concentration of Chemical Compounds

**DOI:** 10.3390/ma14061375

**Published:** 2021-03-12

**Authors:** Marcin Gołąbczak, Andrzej Gołąbczak, Barbara Tomczyk

**Affiliations:** 1Department of Production Engineering, Institute of Machine Tools and Production Engineering, Lodz University of Technology, Stefanowskiego Str. 1/15, 90-924 Łódź, Poland; 2Department of Management Engineering, Institute of Social and Technical Sciences, State Vocational University in Włocławek, 3 Maja Str. 17, 87-800 Włocławek, Poland; andrzej.golabczak@puz.wloclawek.pl; 3Department of Mechanics and Building Structures, Institute of Civil Engineering, Warsaw University of Life Sciences, Nowoursynowska Str. 166, 02-787 Warszawa, Poland; barbara_tomczyk@sggw.edu.pl

**Keywords:** superhard grinding wheels, electrochemical dressing, operational properties, ecological electrolytes, electrochemical reactions, erosion products

## Abstract

This article introduces significant cognitive and usable values in the field of abrasive technology especially in the development of new methods of the electrochemical dressing of superhard grinding wheels with metal bonds. Cognitive values mainly concern the elaboration of the theoretical backgrounds of the electrochemical digestion of compounds of grinding wheel metal bond and gumming up products of the cutting surface of grinding wheel (CSGW). Cognitive values also deal with determining the mathematical relationships describing the influence of technological conditions of dressing on shaping of cutting abilities of superhard grinding wheels. On the other hand, the useful values refer to the industry implementation of the elaborated method and equipment for the electrochemical dressing of suparhard grinding wheels using alternating current (ECDGW-AC). The cost of the device for the realization of this process is low and can be applied in the production conditions. The novel achievements presented in the article are: the elaboration of a new method and equipment for electrochemical dressing of superhard grinding wheels (ECDGW-AC), the selection of electrolytes of low concentration of chemical compounds, tests concerning the digestion of grinding wheel metal bond compounds and gumming up products of CSGW using X-ray analysis, as well as the determination of chemical reactions taking place during elaborated new dressing process, the elaboration of mathematical relationships describing influence of technological conditions of this process on dressing speed and shaping of cutting abilities of superhard grinding wheels, and the performance of technological tests of dressing of superhard grinding wheels using ECDGW-AC method. The elaborated method can be used in ambient temperature and does not cause thermal damages of abrasive grains of cutting surface of grinding wheel and is useful not only for dressing super hard grinding wheels but also for correcting their geometrical deviations.

## 1. Introduction

Metal bonded grinding wheels made of superhard abrasive materials (e.g., natural crystal diamond-CD and synthetic diamond-SD or cubic boron nitride-CBN) fulfill significantly modern requirements concerning grinding processes challenges, especially the precise machining of hard metal alloys or composites [1,2,3].

The rational and economical exploitation of these kind of grinding wheels is conditioned by overcoming the difficulties in shaping their macro- and microgeometry in the sharpening process, both for the manufacturers and users of grinding wheels. These difficulties mainly result from specific physico-mechanical properties of the grinding wheels including the especially high hardness of abrasive material and metal bond strength [4,5].

For this reason, the use of conventional methods for dressing of superhard grinding wheels, e.g., dressing with diamond single-grain dressers, diamond rotating dressers, kneading with a hard roller, etc., is ineffective and uneconomical [1,6,7].

The wear of grinding wheels in the process of grinding with superhard grinding wheels occurs as a result of the phenomenon of gumming up of the cutting surface of the grinding wheel (CSGW) by grinding products and abrasive wear of abrasive grains [2,8,9,10]. In order to counteract the occurrence of the grinding wheel gumming up, the grinding wheel is impregnated with molybdenum compounds (MoS_2_) [11,12]. However, the wear of the abrasive grains of the grinding wheel requires their removal from the CSGW in the process of dressing with erosive methods. The analyses of the source publications and the authors’ investigation results indicate the significant role of erosive methods of superhard grinding wheels dressing in solving the difficulties in forming the cutting abilities of superhard grinding wheels, e.g.,: electrochemical dressing [13,14,15], electrolytic in-process dressing—ELID [16,17], electrochemical dressing with foil electrodes [18], electrodischarge dressing [19,20,21], electrodischarge and electrochemical dressing [13,22,23,24], and laser dressing [25]. Among the erosive methods for dressing superhard grinding wheels there are very attractive and effective electrochemical methods based on the application of alternating current (ECDGW-AC) [22,23,24,26,27,28]. However, the disadvantage of the electrochemical methods is the necessity of using high concentration electrolyte solutions of chemical compounds (15 ÷ 40%).

The presence of electrolytes solutions creates a significant nuisance value for the environment and surroundings, including the corrosive effect of electrolytes on grinder assembly, harmfulness for grinder operators, ecological limitations while utilizing used electrolytes solutions, and challenges arising from grinding hazardous waste [22,24,28,29].

The level of nuisance of electrolytes solutions depends on the chemical composition and concentration of chemical compounds. In order to limit these nuisances, we have analyzed the theoretical literature and carried out the following range of experimental tests: the selection of non-toxic chemical compounds in electrolytes solutions, the reduction of concentration of these compounds in electrolyte, and the application of protective additives reducing corrosivity of grinder assembly.

## 2. Materials and Methods

### 2.1. Main Aim of the Investigations

The aims of the investigations carried out were the elaboration of a new method and equipment for the electrochemical dressing of superhard grinding wheels (ECDGW-AC), the selection of chemical compound of low concentrated electrolytes solutions, the determination of electrochemical reaction in process of anodic digestion of components of metal bond of grinding wheel and grinding materials, the X-ray identification (TUR-M62, Carl Zeiss AG, Jena, Germany) of products arising in electrochemical erosion, and an assessment of the effectiveness of grinding wheels dressings. In the following chapter, detailed information concerning experimental conditions and examples of the investigation results are presented.

### 2.2. Characteristics of the Abrasive Tools and the Machine Tool

For the investigations, the precision universal tool grinding machine Tacchella 4AM (Pabianice, Poland) and the diamond superhard grinding wheels with a metal bond of technical designation S3020 175 × 6 × 3 × 50 D125/100 M100 (“VIS” S.A., Warszawa, Poland) were used. Their main characteristics are depicted in Table 1.

### 2.3. Electrochemical Method of Grinding Wheels Dressing Using Alternating Current

The electrochemical method for superhard grinding wheels dressing based on applications of an alternating current has been proposed in [3,4,6]. The presence of specially selected electrolytes with low concentration of chemical compounds (about 5%) is required in this method.

The dressing process of the face superhard grinding wheel consists of the anodic digestion of the metal bond—5 and gumming up products of cutting surface of superhard grinding wheel (CSGW)—6, with the use of alternating current applied from the current power supply—1 to two carbon electrodes—2. The digestion process of the face superhard grinding wheel (metal bond and gumming up products CSGW) proceeds in the inter-electrode gap—*S*, to which the low concentrated electrolyte solution—*ES* is supplied under pressure of specially selected chemical compound (Figure 1).

The process of anodic digestion of the grinding wheel compounds has a cyclical nature because of the alternating current change conditions of the grinding wheel polarization, which alternately is the anode or cathode, adequate to the variation frequency of the alternating current (Figure 2).

The digestion of the grinding wheel compounds takes place during positive polarization of the grinding wheel (when it is the anode). During the negative periods of the grinding wheel (when it is the cathode), the so-called cathode polarization of the grinding wheel takes place. This polarization favors tearing off the passive layer from the CSGW (metal oxides and other chemical compounds) arising during anodic process. The removal of the passive layer from the CSGW is a favorable process because it increases polarization resistance in the interelectrode gap which slows down the digestion speed of grinding wheel compounds and decreases the dressing efficiency. The main advantages of the ECDGW-AC method are the commutator-free power supply to the grinding wheel, lower cost of power supply unit, cathode activation of the grinding wheel subjected to dressing, application of diluted solutions of electrolytes, and displaying insignificant noxiousness for the surroundings and the natural environment.

The work stand constructed for the investigations consists of an appropriately adapted tool grinder equipped with the electrochemical attachment and systems enabling control and measurements (Figure 3).

The stand provides sharpening of grinding wheels using either stationary electrode (and variable spacing of electrodes) or an electrode with a step feed (providing stabilization of the spacing of electrodes). The linear grinding wheel decrement (*ΔH*) was determined using the contactless inductive sensor (4) equipped with LC generator (11). The stabilization of the inter-electrode gap (*S*) was provided by the microcomputer regulator (12) by means of the stepping motor (9) and the guide-screw (8) feed—in the set of electrodes (7) to the grinding wheel (1). Basic parameters and results of the process were registered with the microcomputer which governed the process according to the special computer software.

### 2.4. Theoretical Analysis of the ECDGW-AC Process

For theoretical considerations, a physical model of the ECDGW-AC process, cf. Figure 1 and Figure 2, has been developed. The following simplifying assumptions have been made for the ECDGW-AC process model [3,4,6]:The grinding wheel digested components (metal bond and products glazing the CSGW) are treated as a homogeneous material.The physical properties of the electrolyte solution in the inter-electrode gap are constant.The distribution of the electrical potential in the inter-electrode gap has a linear character.There are no hydrodynamic interfering phenomena of continuity of electrolyte flow in the inter-electrode gap.The total polarity of the electrodes does not depend on the current density.

Taking into account the assumptions given above, it may be stated that Ohm’s law is satisfied in the ECDGW-AC process, while the current density (*i*) on the anode is defined by the following equation [22]:(1)$$i=\chi \frac{U_{e}-E}{S}$$
where *χ*—the specific conductance of the electrolyte, *U_e_*—the inter-electrode voltage, *E*—the total overpotential, *S*—the thickness of the inter-electrode gap.

For the ECDGW-AC process, the model using a stationary electrode (with a variable thickness of the inter-electrode gap) and the rate of digestion of the grinding wheel components is described by the following differential equation:(2)$$\frac{dS}{dt}=\chi k_{v}\frac{U_{e}sin\omega t-E}{S}$$
where *k_v_*—the coefficient of electrochemical workability of the grinding wheel components, *U_e_*—the effective value voltage of the inter-electrode voltage.

To determine the thickness of the inter-electrode gap in the period *t*ϑ < 0, *T*/2 corresponding to “the positive semi-wave” of the working voltage (Figure 4), Equation (2) must be integrated in the interval *t_i_* < *t* < 1/2*T* − *t_i_*, with an initial condition *S*(*t* = 0) = *S*_0_.

The integral equation takes the following form:(3)$$\overset{S}{\underset{S_{0}}{\int }}SdS=\overset{1/2T}{\underset{0}{\int }}k_{v}\chi \left(U_{m}sin\omega t-E\right)dt-2\overset{t_{i}}{\underset{0}{\int }}k_{v}\chi \left(U_{m}sin\omega t-E\right)dt$$

Determining constants *D* and *C_E_*:(4)$$D=U_{m}k_{v}\chi ,\;\;\;\;C_{E}=\frac{1}{\pi }sin^{2}\left(\frac{1}{2}arcsin\frac{E}{U_{m}}\right)-\frac{E}{2\pi U_{m}}arcsin\frac{E}{U_{m}}$$

The solution to Equation (3) leads to the following equation:(5)$$S_{1}^{2}-S_{0}^{2}=2DT\left(\frac{1}{\pi }-\frac{E}{4U_{m}}-2C_{E}\right)$$

The Equation (5) describes a change in the inter-electrode gap thickness. It can be seen that the elementary layers of the grinding wheel will be digested for successive semi-waves of the working voltage. Thus, according to relationship (5), a change in the inter-electrode gap thickness is described by a series of Equation (6):(6)$$S_{n}^{2}-S_{0}^{2}=2DT_{n}\left(\frac{1}{\pi }-\frac{E}{4U_{m}}-C_{E}\right)$$
where $T_{n}$—the time of dressing needed for the removal of the grinding wheel layer on the part of its circumference corresponding to the width of the tool electrode *b*. By introducing an effective time of dressing $t_{oe}$, determined by Equation (7) and following transformations, Equation (8) defining the inter-electrode gap thickness was obtained:(7)$$t_{oe}=t_{o}\frac{b}{\pi D}$$
where $t_{o}$—the actual time of dressing, *D*—the grinding wheel diameter.
(8)$$S_{n}=\sqrt{S_{0}^{2}-2D\left(\frac{1}{\pi }-\frac{E}{4U_{m}}-2C_{e}\right)t_{oe}}$$

Since in the ECDGW-AC process the inequality $E\ll U_{m}$ is satisfied, Equation (8) can be reduced to the following simpler form:(9)$$S_{n}=\sqrt{S_{0}^{2}-2Dt_{oe}}$$

The thickness of the abrasive layer removed *Δ**H* in the ECDGW-AC process is defined by:(10)$$\Delta H=S_{n}-S_{0}=\sqrt{S_{0}^{2}-2Dt_{oe}}-S_{0}$$

The effective time of dressing determining the technological removal of the digested layer of the grinding wheel *Δ**H* is defined by:(11)$$t_{oe}=\frac{\Delta H\left(\Delta H+2S_{0}\right)}{2D}$$

For the model of the ECDGW-AC process making use of the continuous feed of the electrode, changes in the inter-electrode gap are compensated by an automatic system of the electrode feed (Figure 5).

Therefore, the inter-electrode gap thickness in the ECDGW-AC process is constant, *V_dig_* = constant. The thickness of the grinding wheel layer removed in the ECDGW-AC process can be determined from the following equation:(12)$$\Delta H=k_{v}\chi \frac{U_{e}}{S_{0}}t_{oe}$$
where *U_e_*—the effective value voltage of the inter-electrode gap.

### 2.5. Selection of Electrolytes Solutions

The selection of chemical compounds and concentration of electrolytes solutions used in the electrochemical dressing of grinding wheel are conditioned by many mutually exclusive criteria. These are, among others: good electrochemical properties of electrolyte, low nuisance for surroundings and environment (e.g.,: in aspect of reprocessing the used electrolytes and dressing products, limitations of corrosivity of the grinding machine units, a fire hazard, safety, and the costs of electrolytes and their shelf life) [22,26,28].

Fulfilling all the above mentioned criteria is a very difficult and complex research task. Therefore, we have focused on selecting electrolytes fulfilling the most important requirements, namely: the reduction of nuisance for surroundings and the environment, limitation of corrosivity of the grinding machine units, and assurance of the effectiveness of the grinding wheels dressing.

Efforts to limit the electrolytes nuisance for surroundings and the environment consist of elimination from their chemical composition toxic and hazardous compounds (e.g.,: nitrides and chromates). Limitations of the corrosivity of the grinding machine units have been achieved by reduction of chemical compounds concentration in solution (to about 5%) and application of the protective additions. Investigations carried out allow for the elaboration of five multicomponent solutions of electrolytes [3,4,14] whose physicochemical properties are presented in Table 2.

## 3. Results and Discussion

### 3.1. Studies on Electrolytes Applied for the ECDGW-AC Process

The investigations concern the determination of electrochemical equilibrium conditions and rates of metal digestion (being components of both metal bond and products loading on the CSGW) in selected solutions of electrolytes, the estimation of minimal density of these metals dissolution current, and an assessment of the corrosive effect of these electrolytes on constructional units of the grinder. The research was based on the digestion of samples of metals making up the metal bond and products loading on the CSGW in 5 different electrolyte solutions E1–E5 (containing up to 5% of the following compounds: NaCl, KNO_2_, KNO_3_, Na_2_CO_3_, CaCO_3_, and a protective agent) and chrono-volt-amperometric measurements. The dissolution of Cu, Sn and Ag, i.e., the components of metal bond of a (S3020 175 × 6 × 3 × 50 D125/100 M100) grinding wheel, as well as Co, Ti and W, which are components of products loading on the CSGW, formed during sintered carbides (type S20S) grinding, was within the scope of the studies. The usefulness of the selected electrolytes was estimated on the basis of the rates of individual metal digestion. The value of the current corresponding to the equilibrium state between the reactions of anodic metal digestion and cathodic reduction of the electrolyte solution components was the factor characterizing the metal dissolution rate in the electrolytes. The values of equilibrium potentials and digestion current were determined based on cyclic volt-amperograms and Tafel’s relationship plots.

The comparative assessment of the effect of individual electrolytes on grinding wheel components dissolution rate is depicted in Figure 6. Their analysis indicates that the chemical composition of electrolyte solutions and the applied protective agent (present in the electrolytes E3 and E4) substantially influence the rate of the grinding wheel components digestion. It is particularly evident in the case of Cu and Sn, which are components of the grinding wheel metal bond, and Co, which binds sintered carbides loading on the CSGW. The ratio of the highest to the lowest rate of these metals digestion in selected electrolytes amounted to almost 3, and the rate was the highest for the E5 electrolyte, followed by the rates of dissolution in E1, E2, E3, and E4. The studies proved that appropriate electrolytes provide digestion of all the metals present either in grinding wheel metal bond (Ag, Cu, Sn) or in products loading on the CSGW (Co, W), that enables selection of the best electrolyte solution, optimal for the ECDGW-AC process actual conditions. The only exception is Ti, which undergoes passivation and forms a layer of oxides on the CSGW. The dissolution of all components of metal bond, grinding wheel and Co, which binds the sintered carbides (WC and TiC) results in the simultaneous removal of Ti oxides from the CSGW within the ECDGW-AC process. This fact was confirmed by X-ray studies on the EDGW–AC process products resulting from electrochemical reactions. The executed chrono-volt-amperometric studies and the analysis of their results revealed that the minimum current density enabling digestion of all the metals being components of the grinding wheel was equal to about 0.3 Acm^−2^ (Figure 7).

#### 3.1.1. Electrochemical Reactions and Erosion Products of the ECDGW-AC Process

The analysis of electrochemical reactions pathways concerned the digestion of all metals, either being components of a grinding wheel (S3020 175 × 6 × 3 × 50 D125/100 M100) metal bond or present in waste materials resulting from sintered carbides (S20S) grinding, found in the CSGW. Experiments on these metals digestion were run in the laboratory scale using an electrolyzer potentiostat/galvanostat (Eco Chemie-Metrohm Autolab B.V., Utrecht, The Netherlands) for measurements and the chrono-volt-amperometric method [3]. The chrono-volt-amperometric plots of these metals digestion in the selected electrolyte solution (E3) are collected in Figure 7.

The curves above present variations in the electrode potentials *E*(*t*) during digestion, corresponding to actual period of grinding wheel sharpening by means of the ECDGW-AC method (about 15 min).

The identification of the erosion products was done using X-ray phase analysis [3]. Examples of the diffraction patterns of the grinding wheel components dissolution are illustrated in Figure 8.

#### 3.1.2. Copper Digestion

The analysis of variations in the electrode potential (the sample) (Figure 7a) indicates that the lowest current density (0.003 Acm^−2^) corresponds to digestion of copper yielding its cations Cu^+^ and Cu^2+^, which react with the solution components. The observation of the electrode surrounding revealed generation of a blue precipitate of CuCl_2_. When the process of digestion was accomplished, the white coating of CuCl on the electrode surface was present. The majority of products slid off the electrode surface and settled on the electrolyzer bottom. After complete sedimentation, these products changed their color to brown. X-ray identification of these products (Figure 8a,b) revealed that on the electrode surface mainly CuCl and 3Cu(OH)_2_CuCl_2_ were formed, and the precipitate from the electrolyte solution (after sedimentation and drying) was mainly composed of Cu_2_O, CuO, and Cu_7_Cl_4_(OH)_10_H_2_O. Taking into consideration the results of the process of anodic dissolution of copper analysis, the following electrochemical reactions were proposed [28]:Cu → Cu^+^ + e     E = +0.279(13)
Cu → Cu^2+^ + 2e     E = +0.093(14)
Cu^2+^ + Cl^−^ + e → CuCl     E = +0.300(15)
Cu^2+^ + 2Cl^−^ → CuCl_2_(16)
Cu + Cl^−^ → CuCl     E = −0.105(17)
2CuCl + H_2_O 21C4 Cu_2_O + 2Cl^−^ + 2H(18)
2Cu_2_O + 4H_2_O + 2Cl^−^ ⇄ 3Cu(OH)_2_CuCl_2_ + 2H^+^ + 4e(19)
E = + 0.207 − 0.0295 pH − 0.0295 (Cl^−^)(20)

The reactions presented above as well as X-ray examinations proved that after relatively short period of time (about 5 h) these reactions products (in the form of precipitate) incubated in the electrolyte solution are converted mainly into copper oxides and the complex compound Cu_7_Cl_4_(OH)_10_H_2_O. This reaction pathway is favorable for the electrolyte solution stability and possibility of its regeneration. A small amount of Cu_7_Cl_4_(OH)_10_H_2_O, detected in the solution, indicated minimally reduced concentrations of Cl^−^ and OH^−^ ions in the electrolyte solution.

#### 3.1.3. Tin Digestion

The analysis of variation in the potential *E*(*t*) (Figure 7b) proves that, already at the lowest digestion current density (*i* = 0.0038 Acm^−2^), Sn^2+^ and Sn^4+^ cations are released. The values of potentials corresponding to this current density are approximately equal to respective standard potentials (for Sn^2+^
*E* = −0.23 V and for Sn^4+^
*E* = 0.382 V). During these reactions, a white, sponge-like coating on the surface of electrode is formed. An increase in the dissolution current density (up to *i* = 0.048 Acm^−2^) results in an increase of the electrode potential and, after polarization time of about 600 s, the electrode is almost dissolved on the boundary of its submerging in the electrolyte solution. White, porous precipitate composed of tin oxides and hydroxides is observed on the electrode surface. The presence of these compounds was confirmed by X-ray analysis (Figure 8c). Taking into consideration these results, the following reactions were proposed [28]:Sn ⇄ Sn^2+^ + 2e     E = −0.382(21)
Sn^2+^ ⇄ Sn^4+^ +2e     E = −0.092(22)
Sn ⇄ Sn^4+^ + 4e     E = −0.232(23)
Sn + 2H_2_O → Sn(OH)_2_ + 2H^+^ + 2e   E = −0.742(24)
Sn + 2H_2_O → SnO_2_ + 4H^+^ + 4e     E = −0.752(25)
Sn + 3OH^−^ ⇄ HSnO_2_^−^ + H_2_O + 2e     E = −1.152(26)
HSnO_2_^−^ + 3OH^−^ + H_2_O ⇄ Sn(OH)_6_^2-^ + 2e E = −1.172(27)
Sn(OH)_2_ → SnO + H_2_O(28)

The results of X-ray examination and the pathway of electrochemical reactions indicate that the process of anodic silver dissolution may lead to a decrease in Cl^−^ ions concentration in the electrolyte solution, because AgCl is the main product of the reactions.

#### 3.1.4. Silver Digestion

The variation in potential *E*(*t*) (Figure 7c) shows that for low current densities (*i* = 0.001 ÷ 0.0025 A cm^−2^), the electrode potentials are constant, and their values correspond to the electrode reaction of silver in the presence of Cl^−^ and OH^−^ ions. In this case, a grey, well-sticking to the electrode surface, AgCl coating was found (chlorine-silver electrode). An enhancement of the current density up to 0.16 A cm^−2^, resulted in an increase of the potential up to 0.55 V, which corresponded to the anodic silver digestion. The absence of silver ions in the electrolyte solution was also revealed. Higher current densities caused a white coating to form, which diffused to the solution. Most probably, it was composed of AgCl or Ag_2_CO_3_. Their presence was confirmed by X-ray analysis of the products of dissolution formed on the electrode surface and the precipitate present in the solution (Figure 8d). The main component of the precipitate formed in the solution appeared to be AgCl together with small amount of AgO. For the analyzed process of anodic silver digestion, the following electrochemical reactions were proposed [28]:Ag ⇄ Ag^+^ + e     E = +0.557(29)
Ag + Cl^−^ ⇄ AgCl     E = −0.200(30)
2Ag + 2OH^−^ ⇄ Ag_2_O+ H_2_O + 2e     E = +0.102(31)
Ag_2_O + 2OH^−^ ⇄ 2AgO + H_2_O + 2e     E = +0.358(32)
2AgO + 2OH^−^ ⇄Ag_2_O_3_ + H_2_O + 2e     E = +0.498(33)
2Ag + CO_3_^2-^ ⇄ Ag_2_CO_3_ + 2e     E = +0.218(34)
Ag_2_CO_3_ → Ag_2_O + CO_2_(35)

The results of X-ray examination and the pathway of electrochemical reactions indicate that the process of anodic silver digestion may lead to a decrease in Cl^−^ ions concentration in the electrolyte solution since AgCl is the main product of the reactions.

#### 3.1.5. Cobalt Digestion

The variation in the cobalt electrode potential *E*(*t*) (Figure 7e) indicates that for a small current density its potential of digestion is negative. The sample’s surface was coated with a pink precipitate, which diffused into the electrolyte solution, thus suggesting the possibility of cobalt hydroxide synthesis. An increase in current density up to *i* = 0.293 Acm^−2^ results in a rapid cobalt conversion to Co^+3^ ions. For this current density, products of digestion were rapidly removed from the sample surface, and after polarization time equal to 500 s, the sudden increase in the potential, accompanied by the complete sample digestion at the borderline of its submerging in the electrolyte solution, were observed. The course of digestion indicates that the products of electrochemical reactions are probably as follows: cobalt chloride, cobalt hydroxide, which is poorly soluble, and cobalt carbonate. An X-ray analysis of dissolution products (Figure 8e) confirmed the presence of the both first compounds, it means CoCl_2_ and Co(OH)_2_. The results of cobalt digestion process permit to anticipate the following reactions [28]:Co ⇄ Co^2+^ + 2e     E = −0.522(36)
Co ⇄ Co^3+^ + 3e     E = +0.088(37)
Co^2+^ ⇄ Co^3+^ + e     E = +1.598(38)
Co + CO_3_^2-^ ⇄ CoCO_3_ + 2e     E = −0.932(39)
Co + 2OH^−^ ⇄ Co(OH)_2_ + 2e     E = −0.972(40)
Co(OH)_2_ + OH^−^ ⇄ Co(OH)_3_ + e     E = −0.0724(41)
Co + 2H_2_O → Co(OH)_2_ + 2H^+^ + 2e     E = −0.562(42)
Co^2+^ + 2Cl^−^ → CoCl_2_(43)
Co^2+^ + C0_3_^2−^ → CoCO_3_(44)

An analysis of the reactions presented above points to a possibility of a decrease in OH^−^ and Cl^−^ ions concentration in the electrolyte solution.

#### 3.1.6. Titanium Carbide Digestion

The variation in potential *E*(*t*) (Figure 7d) suggests that the oxide layer is formed on the electrode surface for the low current density. This conclusion is confirmed by a white coating on the surface of the sample. The presence of titanium oxide was also detected using X-ray analysis (Figure 8f).

Digestion of this compound is based on its decomposition [28]:TiC + 6OH^−^ → TiO_2_ + CO_2_ + 2H^+^ + 2H_2_O + 8e(45)

The reactions leading to titanium oxides synthesis during the process of dissolution are probably as follows [28]:Ti ⇄ Ti^2+^ + 2e     E = −1.862(46)
Ti^2+^ ⇄ Ti^3+^ + e     E = −0.612(47)
Ti + 2H_2_O ⇄ TiO_2_ + 4H^+^ + 4e     E = −1.102(48)
Ti + H_2_O ⇄ TiO + 2H^+^ + 2e     E = −1.962(49)
Ti + H_2_O ⇄ TiO^2+^ + 2H^+^ + 4e     E = −1.122(50)
Ti^3+^ + H_2_O ⇄ TiO^2+^ + 2H^+^ + e     E = −0.142(51)

Titanium oxides point to the passivation of this metal. Because its ions are generated at negative potential values, it is likely that they are present in the electrolyte solution.

#### 3.1.7. Tungsten Carbide Digestion

The variation in potential *E*(*t*) (Figure 7f) indicates that for low current densities, the electrode potential is negative. Its value equal to −0.25 V corresponds to the reaction of tungsten ionization. An increase in the current density results in the passivation of the sample’s surface, analogously to titanium. The dissolving of tungsten carbide leads to its decomposition to tungsten acid and carbon dioxide, according to the following equation [28]:WC + 10OH^−^ → WO_4_^2−^ + CO_2_ + 2H^+^ + 4H_2_O + 10e(52)

During the process of tungsten dissolution, a dark coating (oxides layer) was observed on the surface of the electrode, together with a relatively high anodic potential (about 2.5 V), appearing already at low current density. This is the evidence of the tungsten passivation during the process. The following tungsten reactions in the applied electrolyte solution are possible [28]:W + 2H_2_O ⇄ WO_2_ + 4H^+^ + 4e     E = −0.362(53)
W + 3H_2_O ⇄ WO_3_ + 6H^+^ + 6e     E = −0.332(54)

The presented reactions prove that tungsten oxides and carbon dioxide are the products of tungsten carbide dissolution.

The investigations revealed that all the metals, both being components of the grinding wheel metal bond (Ag, Cu, Sn) and cobalt, which is the binding metal for sintered carbides (TiC–Co and WC–Co), loading on the CSGW, undergo the anodic dissolution. Dissolution of cobalt results in decomposition of the carbides TiC and WC, mainly to CO_2_ and metal oxides, which are passive against the solution. However, their passivity has no deleterious effect on the grinding wheel sharpening, since dissolution of the metals-components of the metal bond (Ag, Cu, Sn) and cobalt, causes their removal from CSGW. The main products of electrochemical reactions corresponding to the dissolution process are chlorides, oxides and carbonates of the dissolved metals. The compounds listed above mainly form easy-to-filter precipitates in the electrolyte solution, which are beneficial for regenerating the solution and utilizing the waste products of the grinding wheels dressing process. It was revealed that the anodic dissolution of metals may be accompanied by a decrease in Cl^−^ and OH^−^ ions concentration in the electrolyte solution. This negative effect may be eliminated with the use of proper chemicals and by controlling the ion concentration and pH of the electrolyte solution.

### 3.2. The Noxiousness of the ECDGW-AC Process for the Surroundings

The process of dressing of superhard grinding wheels using the ECDGW-AC method requires the application of electrolytes that are noxious for the surroundings and the natural environment. Their negative effect consists, among other things, of the corrosive action on the grinding machine units and the noxiousness for the operators of the grinding machines and for the surroundings. There are also ecological restrictions and difficulties related to the utilization of dressing waste and regeneration of electrolytes. The degree of the problems mentioned above depends mainly on the chemical composition of the chemical compounds and their concentration in the electrolyte. The investigations carried out confirmed the possibility to limit this noxiousness in the process of grinding wheel dressing especially by eliminating toxic compounds from the electrolyte composition (e.g., nitrites and chromates) and reducing the chemical compound concentration to about 5%. The corrosive action of electrolytes on the grinding wheel units was limited by the use of electrolytes of a high pH of 9.4–12.5, the selection of their chemical composition. and the use of protective agents. Examples of the results of the laboratory investigations into the rate of corrosion (determined by the single-sweep voltammetry method) of steel sheet 815 in different electrolytes are shown in Figure 9.

The slowest rate of corrosion was obtained for electrolytes E4 and E3, which contained a protective agent. The effect of the remaining electrolytes, different composition of chemical compounds can be ordered as follows: E2, E1, and E5. It should be noted that the use of protective agents can also attenuate the effectiveness of electrolytes in the dressing process. Therefore, these two processes should be analyzed simultaneously. Also, the investigations concerning the description of electrochemical reactions and identification of the products formed in the dressing process are continued. Their knowledge is of great significance in view of the requirements of the environmental protection and the possibility of electrolyte regeneration.

### 3.3. Experimental Investigations of the ECDGW-AC Process

The comprehensive results of the experimental studies on the ECDGW-AC process and the description of the test stand contain the authors’ earlier publications [4,6]. The paper sample presented the results of the experimental investigations aiming at the verification of the mathematical relationships established and the assessment of the effectiveness of the ECDGW-AC process using a stationary tool-electrode and, comparatively, with its feed stabilizing the inter-electrode gap thickness in the ECDGW-AC process. The automatic feed of the tool electrode is affected by a control system whose diagram is shown in Figure 5. An inductive contactless sensor L interacting with a frequency generator LC was used for the measurement of the inter-electrode gap thickness. The tool electrode feed is affected by a step motor controlled by a computer.

The characteristic results of the investigations of the ECDGW-AC process are presented in Figure 10.

They refer to the comparison of the theoretical calculations of the dissolution rate of the cutting abrasive layer *(Δ**H*) with the results of the experimental investigations. This comparison of the investigation results confirms the significant effect of the tool electrode feed on the effectiveness of the ECDGW-AC process. The tests of dressing of a grinding wheel (S3020 175 × 6 × 3 × 50 D125/100 M100) with the feed of the tool electrode have shown an increase in the effectiveness of the dressing process by about 45% compared with the dressing process with a stationary electrode. The application of the feed of the tool electrode in the ECDGW-AC process has a favorable effect on the reduction of losses of the dissolution current and stability of the dressing process [28]. The investigations carried out have also confirmed the good conformity of the theoretical calculations with the experimental investigations. The differences in the rate of digestion did not exceed 50% in the case of dressing with a stationary electrode and about 20% when the feed of the electrode was used, which testifies the correct development of the ECDGW-AC process. The examples of experimental data confirming the possibility of reduction in axial run—out of a face of a grinding wheel in the ECDGW-AC process are shown in Figure 11.

The comparison of experimental and theoretical values of the coefficient of reduction in deviation of the run—out of a grinding wheel in the ECDGW-AC dressing process. They refer to dressing of a diamond grinding wheel of the known initial value of face run-out *ε**_O_*. The dressing process was carried out with the electrode as the stationary tool. In the course of the dressing, the face run-out *ε*(*t*) and the thickness of the removed layer of the grinding wheel (*Δ**H*) were determined.

The analysis of the results confirms that reduction of the deviation (*ε**_O_*) in the ECDGW-AC process is possible and points to conformity of the theoretical calculations with experimental data.

The ECDGW-AC process effectiveness in the forming of the macro-and microgeometry cutting surface grinding wheel (CSGW) was assessed by stereometric parameters, such as: the static cutting edges (*S_N_*), the kinematic cutting edges (*H_N_*), and the average maximum thickness of a chip (*T_N_*). They were determined using the computer analysis of circumferential representation of the CSGW profile and simulation of the grinding process. Peaks of the CSGW profile, of which the minimum height was equal to 1 μm, and which were present at a depth up to 30 μm from the nominal surface CSGW surface, were identified as the static cutting edges (*S_N_*) [28]. Examples of the results of the investigations illustrating the effect of the ECDGW-AC process on the formation of the CSGW macro- and micro-geometry are presented in Figure 12.

This is the comparison of the stereometric parameters of the CSGW for different conditions of the grinding wheel (S3020 175 × 6 × 3 × 50 D125/100 M100), it means for the grinding wheel glazed in the test of grinding of sintered carbides (S20S) and further subjected to dressing by the ECDGW-AC method. The static edges (*S_N_*) were assessed basing on the changes in their total number (Σ*S_N_*) and the average depth of deposition ($\bar{Z}$) in the CSGW (Figure 12a). The most important changes in these parameters were observed in the initial period of dressing, in which the removal of the wheel layer (Σ*H_N_*) is approximately 20 μm (Figure 12b). On the completion of the ECDGW-AC process, very favorable changes in these parameters were obtained: a decrease in Σ*S_N_* by approximately 46% and an increase in the depth of deposition of the static edges by approximately 140%.

The kinematic cutting edges *H_N_* (Figure 12b) were identified from a set of static edges basing on the coefficient of “simulation of the grinding process”—*k*, which is the condition of penetration of the CSGW into the workpiece [28]:(55)$$k=2\frac{V_{f}}{V_{GW}}\sqrt{\frac{a}{D_{GW}}}$$
where *V_GW_*—speed of grinding wheel, *V_f_*—speed of longitudinal feed, *a*—depth of grinding, *D_GW_*—equivalent grinding wheel diameter.

Under the experimental grinding conditions [28]: *k* = 0.712 × 10^−4^. An analysis of the investigation results shows that the tendency of changes in Σ*S_N_* (Figure 12a) and *Z* in the CSGW is similar to that of the static edges, since by using the ECDGW-AC process a decrease in the number of kinematic edges by approximately 51% and an increase in the depth of their deposition in the CSGW by approximately 85% was obtained. This favorable direction of changes in the Σ*S_N_* and z parameters confirms the effectiveness of the ECDGW-AC process in restoring the cutting ability of the grinding wheel subjected to dressing. The average maximum thickness of chips (*T_N_*) is determined by the kinematic edges (*H_N_*) and distance between them (*L_N_*), according to the equation:(56)$$T_{N}=kL_{N}-\left(H_{N}-H_{N-1}\right)$$
where *L_N_*—distance between kinematic edges.

An analysis of the changes in this parameter (Figure 12) confirms the effectiveness of the ECDGW-AC process in forming the CSGW, particularly in restoring the cutting ability of the grinding wheel. Following the dressing by ECDGW-AC method, a change in this parameter is observed consisting of the histogram shift of distribution of *T_N_* into a zone of greater dimensions of the chip. This attests to an increase in the proportion of the cutting edges actively participating in chip formation and a limitation of the number of apparently active cutting edges. The loading of grinding products on the CSGW is one of the forms of superhard grinding wheels wear. An X-ray method of the modified external standard, developed by the author, was used in order to assess the degree of loading the CSGW with grinding products. Some examples of investigation results, confirming the effectiveness of the ECDGW-AC process for the removal of products loading on the CSGW of the (S3020 175 × 6 × 3 × 50 D125/100 M100) grinding wheel, glazed in test of sintered carbides (S20S) grinding, are shown in Figure 13.

Their analysis indicates that the most intensive removal of products loading on the CSGW occurs in the initial period of dressing, approximately 2 min. The complete removal is obtained after 5 to 6 min of dressing. A layer of the grinding wheel of an approximate thickness 10 µm is removed over this dressing period. The results confirm the effectiveness of the ECDGW-AC process in the removal of grinding products gumming up the CSGW. 

## 4. Conclusions

The presented investigation results confirm the possibility of the application of superhard grinding wheels (ECDGW-AC) solutions of low concentration electrolytes of chemical compounds during the electrochemical dressing process. The determined chemical composition of low concentration solutions of electrolytes (containing about 5% of chemical compounds) assures the required efficiency of the dressing process of grinding wheels using ECDGW-AC method. Solutions of electrolytes are characterized by lower nuisance for surroundings and the environment and low corrosivity of the grinding machine units. Electrochemical studies allow for the elaboration of the course of electrochemical reactions during ECDGW-AC process. However, X-ray investigations can be used for the precise identification of the erosion products. It is demonstrated that an anodic dissolution is observed for all metals composing the metal bond of grinding wheel and grinded materials.

The main products of coursed electrochemical reactions are chlorides, oxides, hydroxides and carbonates of digestion metals. The above mentioned chemical compounds are present in solutions of electrolytes as sediments which can be easily filtered out. Their chemical analysis does not reveal the presence of chemical compounds which can be harmful to the environment or for the operators of the grinding machines Investigation results confirm possibility of reduction of corrosive influence of solutions of electrolytes on the corrosivity of the grinding machine units by application of alkali solutions of pH ≥ 9.4 and protective additives.

The theoretical consideration and the experimental investigations presented here have shown a significant effect of the feed of the tool–electrode on the effectiveness of grinding wheel dressing in the ECDGW-AC process. The application of the proper feed of the tool electrode leads to reduction of the dissolution current losses. Moreover, the efficiency of the dressing process increases. 

The ECDGW-AC method is particularly predisposed for forming of the CSGW macro- and micro-geometry, removing the grinding products loading on the CSGW, and for correcting the CSGW profile. The investigations have confirmed that this kind of dressing method can be recommended for dressing of superhard grinding wheel. The implementation of the ECDGW-AC method requires equipping the grinding machine with a cheap and simple electrochemical unit.

Currently, there are realized industrial tests concerning the implementation elaborated dressing method of superhard grinding wheels (ECDGW-AC) in factory producing abrasive tools. The elaborated method of electrochemical dressing of superhard grinding wheels (ECDGW-AC) has received polish patents no PL 169942 and PL 190236 [30,31].

## Figures and Tables

**Figure 1 materials-14-01375-f001:**
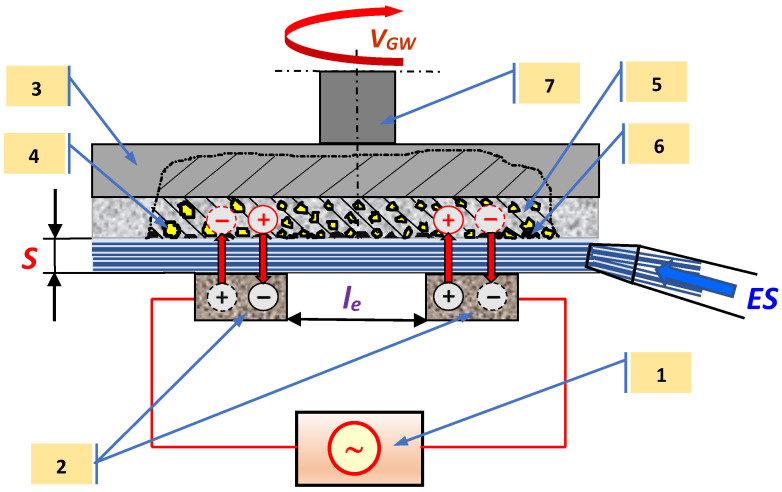
The scheme of the electrochemical dressing of grinding wheel using alternating current (ECDGW-AC): 1—alternating current power supply, 2—carbon electrodes, 3—grinding wheel frame, 4—abrasive grains, 5—metal bond, 6—gumming up products of cutting surface of superhard grinding wheel (CSGW), 7—spindle; *V_GW_*—speed of grinding wheel, *S*—inter-electrode gap, *l_e_*—distance between electrodes, *ES*—electrolyte solution of specially selected chemical compounds.

**Figure 2 materials-14-01375-f002:**
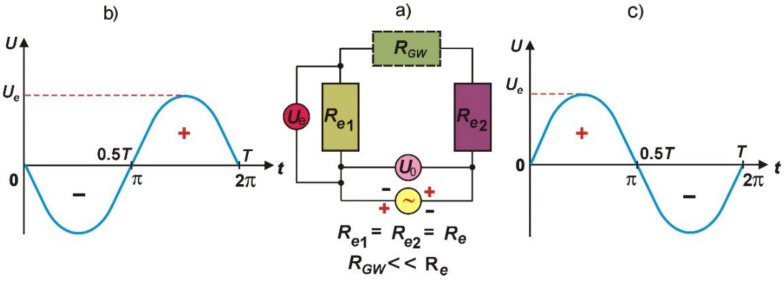
(**a**) Equivalent electric circuit diagram of the dressing process using ECDGW-AC method; (**b**,**c**) Course of current variation on electrodes; *R_e_*—resistance of electrolyte, *R_GW_*—resistance of grinding wheel.

**Figure 3 materials-14-01375-f003:**
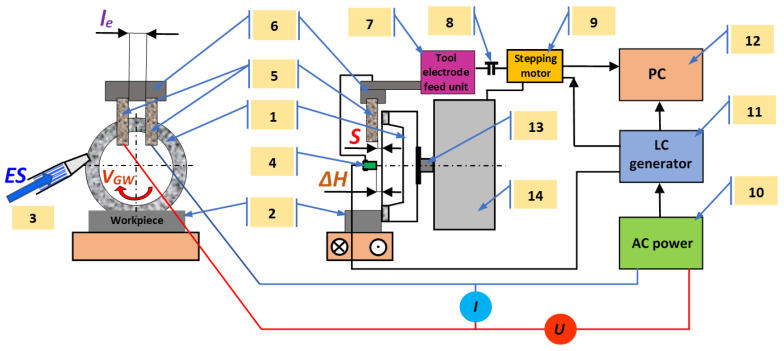
The scheme of a stand for grinding wheels dressing using the ECDGW-AC method: 1—grinding wheel, 2—workpiece, 3—electrolyte supplying nozzle, 4—inductive sensor, 5—tool-electrode, 6—electrode holder, 7—tool-electrode feed unit, 8—guide-screw, 9—stepping motor, 10—AC power unit, 11—LC generator, 12—computer, 13—spindle, 14—headstock, *l_e_*—distance between electrodes, *ES*—electrolyte solution of specially selected chemical compounds, *S*—inter-electrode gap, *ΔH*—the thickness of the grinding wheel layer removed in the ECDGW-AC process.

**Figure 4 materials-14-01375-f004:**
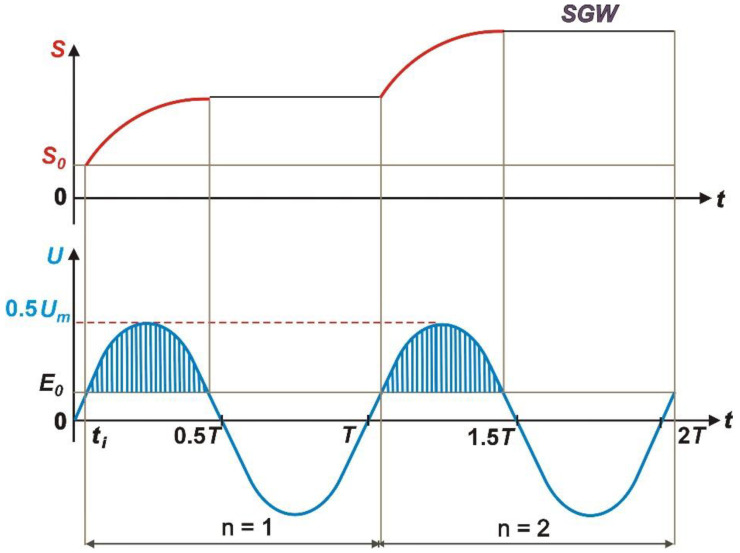
A curve of the inter-electrode voltage *U_m_* and inter inter-electrode gap during ECDGW-AC process, *SGW*—surface of grinding wheel, *t_i_*—the time of start of digestion process.

**Figure 5 materials-14-01375-f005:**
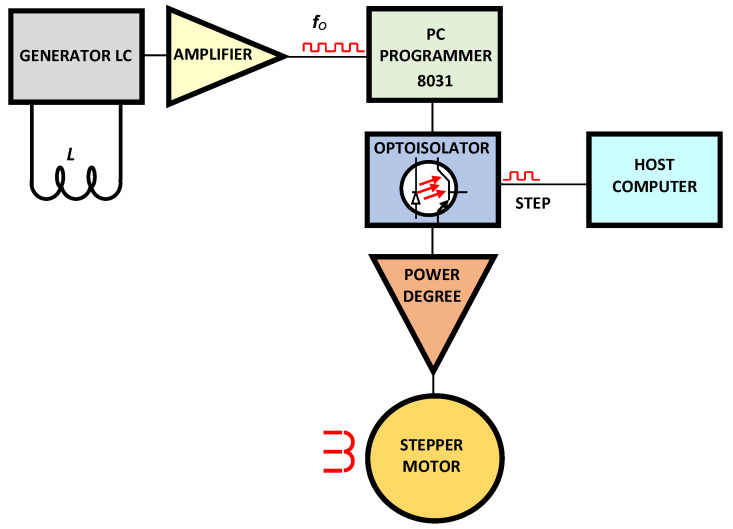
A schematic diagram of system for control of tool-electrode feed.

**Figure 6 materials-14-01375-f006:**
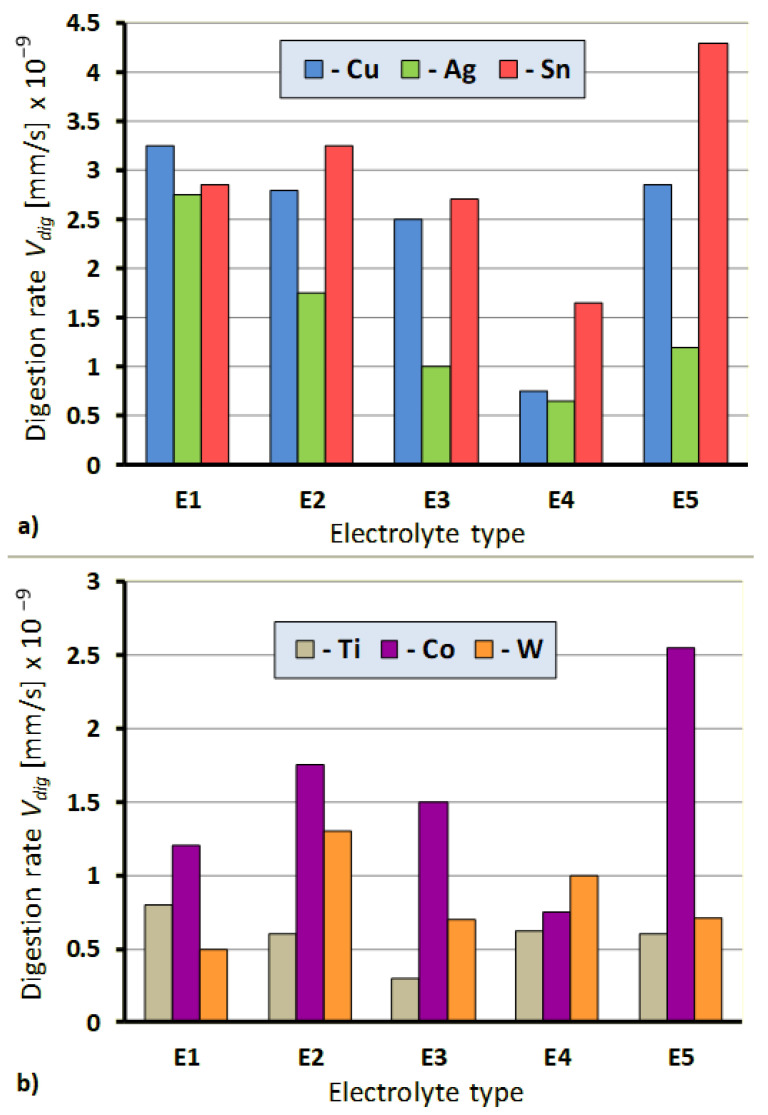
The effect of electrolyte composition (Table 2) on the digestion rate of the grinding wheel components: (**a**) Grinding wheel metals bond; (**b**) Metals in products of loading in the cutting surface of grinding wheel (CSGW).

**Figure 7 materials-14-01375-f007:**
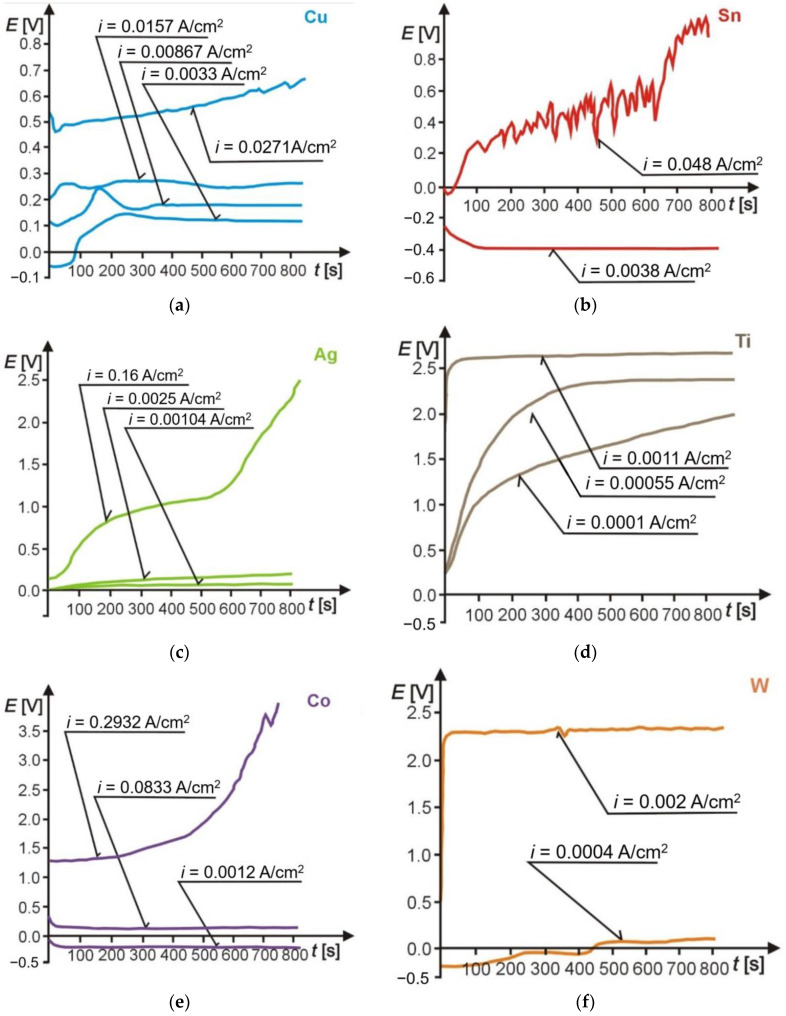
The chrono-volt-amperometric plots of the grinding wheel components: (**a**) Copper; (**b**) Tin; (**c**) Silver; (**d**) Titanium; (**e**) Cobalt; (**f**) Tungsten.

**Figure 8 materials-14-01375-f008:**
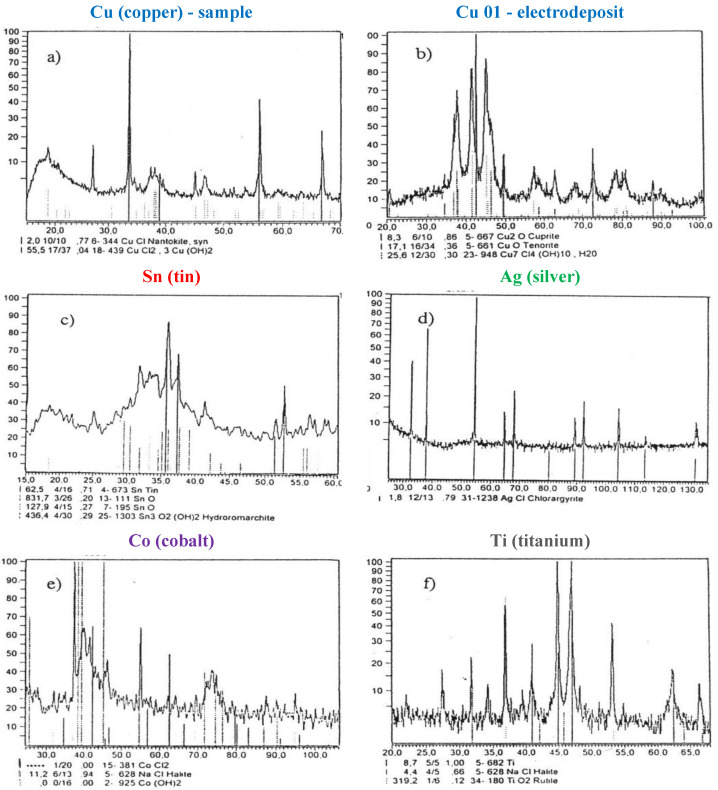
X-ray diffraction patterns of the grinding wheel erosion products: (**a**) Copper—the sample; (**b**) Copper—electrodeposit; (**c**) Tin; (**d**) Silver; (**e**) Cobalt; (**f**) Titanium; Electrolyte solution E3.

**Figure 9 materials-14-01375-f009:**
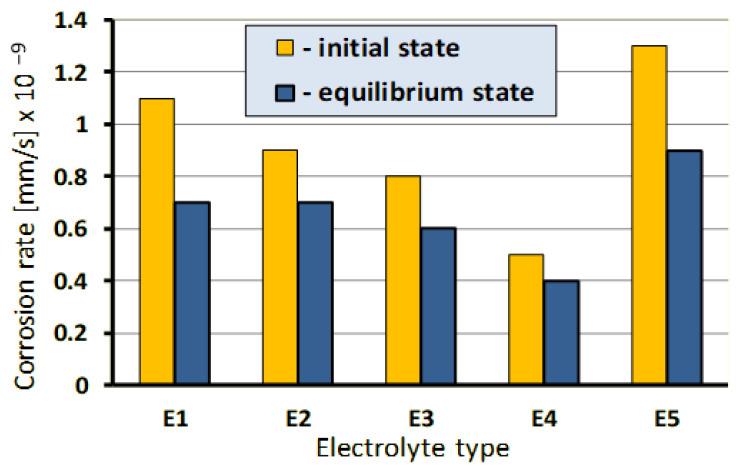
The effect of the electrolyte composition on the corrosion rate of constructional steel; Electrolytes as in Table 2.

**Figure 10 materials-14-01375-f010:**
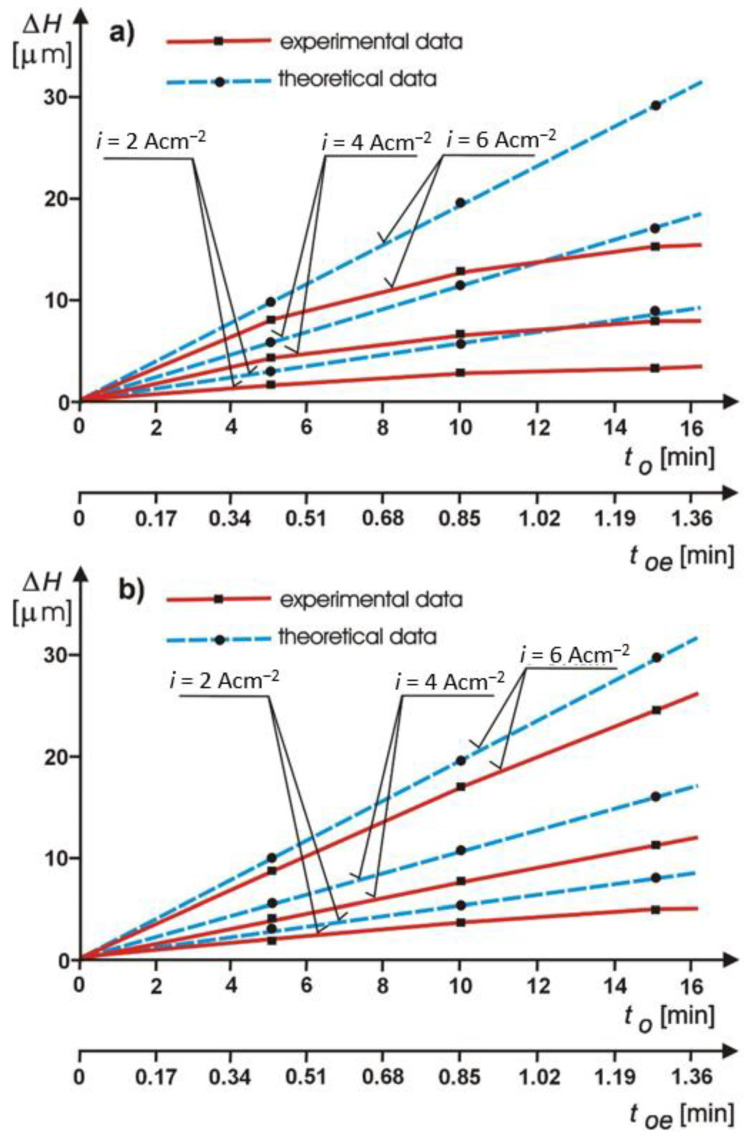
A comparison of the theoretical calculations and the experimental investigations of the grinding layer *ΔH* removed in the ECDGW-AC dressing process: (**a**) Dressing with a stationary electrode; (**b**) Dressing with a step feed electrode.

**Figure 11 materials-14-01375-f011:**
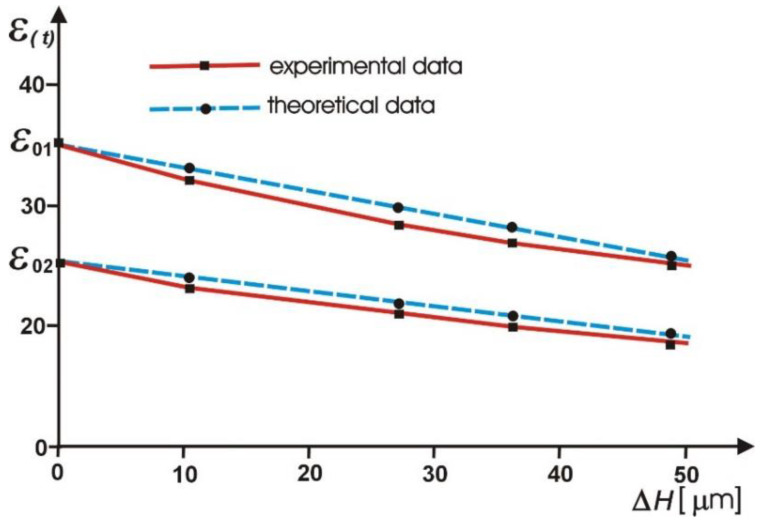
The comparison of experimental and theoretical values of the reduction of the run—out deviation (*ε**_O_*) of a grinding wheel in the ECDGW-AC process.

**Figure 12 materials-14-01375-f012:**
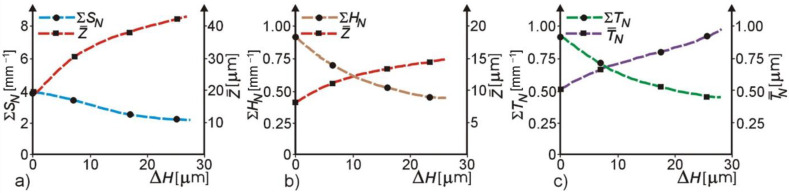
The stereometric parameters CSGW in ECDGW-AC process: (**a**) Static cutting edges; (**b**) Kinematic cutting edges; (**c**) Maximum chip thickness; Grinding conditions: *V_GW_* = 25 m/s, *V_f_* = 5 m/min, *a* = 0.02 mm; Dressing conditions: *V_GW_* = 25 m/s, *i* = 6 A/cm^2^, *U_e_* = 18.7 V.

**Figure 13 materials-14-01375-f013:**
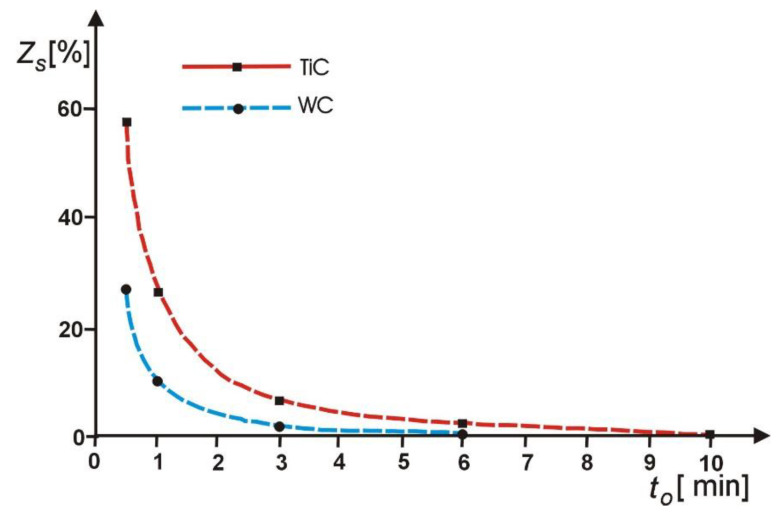
Effect of dressing time on the degree of gumming up products of cutting surface of superhard grinding wheel (CSGW).

**Table 1 materials-14-01375-t001:** Characteristics of the grinding wheels used in the investigations.

Technical Designation	S3020 175 × 6 × 3 × 50 D125/100 M100
Manufacturer	“VIS” S.A. (Warszawa, Poland)
Type of Grinding Wheel	Flat diamond cup grinding wheel
Dimensions	External diameter D = 125 mm, inner diameter d = 50 mm, abrasive layer width w = 6 mm, thickness of abrasive layer x = 3 mm
Abrasive Grain Type	Monocrystalline synthetic diamond
Abrasive Grain Fracture Number	125
Bond	Metal
Grains Concentration	100
Grinding Machine	Tacchella 4AM fum Ponar-Pabianice (Poland)

**Table 2 materials-14-01375-t002:** Physicochemical properties of electrolytes solutions.

**No.**	**Chemical Composition and** **Electrolyte Properties**	**Symbol of Electrolyte Solution (ES)**				
		**E1**	**E2**	**E3**	**E4**	**E5**
1.	chemical composition	NaCl, KNO_2_, KNO_2_, KNO_3_, CaCO_3_	NaCl, KNO_2_, KNO_2_, KNO_3_, Na_2_CO_3_	NaCl, KNO_3_, Na_2_O_3_,	NaCl, KNO_2_, KNO_2_, KNO_3_, Na_2_CO_3_	NaCl, KNO_3_, Na_2_O_3_
2.	concentration of solution [%]	5	5	5	5	5
3.	protective addition [%]	-	-	0.5	1	-
4.	density of electrolyte [g/cm^3^]	1.04	1.039	1.039	1.042	1.043
5.	pH of electrolyte	9.7	10.81	10.80	10.81	10.90

## Data Availability

Data is contained within the article.

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
