# Peer review of "Electrochemical and X-ray Examinations of Erosion Products during Dressing of Superhard Grinding Wheels Using Alternating Current and Ecological Electrolytes of Low Concentration of Chemical Compounds"

_materials, 2021, doi:10.3390/ma14061375_

Round 1

Reviewer 1 Report

The reviewer comments of the paper «Method of Electrochemical Dressing of Superhard Grinding Wheels Using Alternating Current and Ecological Electrolytes of Low Concentration of Chemical Compounds»- Reviewer

The authors presented an article «Method of Electrochemical Dressing of Superhard Grinding Wheels Using Alternating Current and Ecological Electrolytes of Low Concentration of Chemical Compounds». However, there are several points in the article that require further explanation.

Comment 1:

Abstract.

Demonstrate in the abstract novelty, practical significance. Add quantitative and qualitative work results to the abstract. Briefly describe the methods used in the research.

Comment 2:

The introduction needs to be significantly improved. Need to add a paragraph to show the relevance of dressing grinding wheels. What is eliminated as a result of this? What are the benefits of dressing grinding wheels? Group citations [1, 5, 10, 18] ... [2, 39 7, 8, 9, 11, 13, 16, 17, 19, 20] do not give any information to the reader. Try to at least break this sentence into several parts: ... [?], ... [?], ... [?]. Or provide a more detailed analysis of each citation.

Add a couple of articles to the introduction:

2019 International Journal of Surface Science and Engineering 2-3 (13): 181-200. doi: 10.1504/IJSURFSE.2019.10024014

2019 Micromachines 10 (4): 255. doi: 10.3390/mi10040255

Each grinding wheel dressing method requires a critical review of the method. And a more detailed analysis of the articles. Also, add at least 10 relevant references published in the last 5 years.

 Before the purpose of the article, "white" spots should be more clearly defined. What has not been previously done and investigated. Based on this, formulate a clearer and more coherent purpose of the article. After the goal, briefly list what has been done in each section of the article.

Comment 3:

  1. Materials and Methods

Add geometric dimensions to the abrasive wheels. What grain size is used. Which bundle is used? Give the designation of the circle according to generally accepted standards. Decipher the parameters. Describe the chemical composition in the grinding wheel parts table.

For devices and machines used in research, indicate in parentheses (manufacturer, city, country).

Comment 4:

  1. Results and Discussion

Are all the formulas in the article original? If not, then appropriate citations are needed.

Be sure to list all physical quantities in it after each formula.

Are all the drawings in the article original? If not needed appropriate citations. And also needs publisher permissions.

Draw figures 4, 6, 9 in color.

Comment 5:

It will be useful to add a section of Nomenclature in which to sign all the physical quantities and abbreviations encountered in the article. There are many physical quantities in the text and such a section will help to find the description of the necessary element.

For example,

S                : the thickness of the inter-electrode gap

CBN         : Cubic boron nitride

etc.

Comment 6:

Conclusions

Add quantitative and qualitative work results. In addition, it is necessary to more clearly show the novelty of the article and the advantages of the proposed method. What is the difference from previous work in this area? Show practical relevance. Conclusions should reflect the purpose of the article.

Comment 7:

The article must be proofread by a native English speaker.

The article is interesting and helpful. However, authors should carefully study the comments and make improvements to the article step by step. Mark all changes in color. After major changes can an article be considered for publication in the "Materials".

Reviewer 2 Report

  1. This publication is interesting, but highly fragmented and long-term "recycled". Above all, the novelty is not explicitly declared. The idea of using "ecological electrolytes" (electrolytes with a low concentration of chemical compounds) is not new for 25 years.
  2. It is presented 56 “new” equations without explicit references to the literature (with the exception of equations (55), (56)). It is necessary to explicitly add which equations are derived as new equations and which equations have already been published in the past.
  3. For continuous reading of the text, it is necessary to renumber citations in the text [1,5,10,18] - [1,2,3,4], [2,3,18] - [4,5,6]), etc.
  4. It is necessary to submit original diagrams, Figs. 1,2,3 are constantly repeated, eg:
  • Golabczak, A.; Golabczak M.: Economical aspect of assessment of electrochemical dressing of super hard grinding wheels. Scientific proceedings international scientific conference "high technologies. business. society 2016" ISSN 1310-3946
  1. Ad 63-64 line: „The electrochemical method of superhard grinding wheels dressing based on applications of an alternating current has been proposed in [3, 4, 6].“ The original issue presented by the author Golabczak, A. is much older, eg:
  • Golabczak, A.: Electrochemical dressing of grinding wheels using alternating current. Scien. Bull. Of Ú Lódž Technical University, No 746, z. 234, Lódž (1996), pp. 159.
  • Golabczak, Andrzej; Kozak, Jerzy; Yamaguchi H.: Study of Electrochemical Dressing of Superhard Grinding Wheels Using Alternating Current. December 1998; In book: Advances in Abrasive Technology II (pp.49-56); Edition: II; Chapter: Study of Elctrochemical Dressing of Superhard Grinding Wheels Using Alternating Current (ECD-AC); Publisher: The Society of Grinding Engineers (SGE), Tokyo, Japan; Editors: Uematsu T., Suzuki K.

Abstract: „This paper presents a method of electrochemical dressing of metal-bonded grinding wheels of superhard materials. Dressing of grinding wheels by this method consists of the anodic dissolution of the compounds of grinding wheel with participation of an alternating current in electolytes of a low concentration of chemical compounds. The method has been discused using theoretical analysis of the electrochemical dressing process and experimental results. The theoretical analysis have been presented based on the mathematical relationships between the parameters of the dressing process, the characteristics of electrolytes and the electrochemically removed thickness of the cutting layer of the grinding wheel. The experimental results demonstrate the theoretical relationships and the usefulness of the dressing method to regenerate and shape functional qualities of superhard grinding wheel.“

  • Golabczak A., Kozak J., Yamaguhi J.: Investigations of the process of electrochemical dressing of grinding wheels using alternating current ACD-AC, Advances in abrasive technology, The Society of Grinding Engineering, Japan, 1999, pp.49-55.
  • Golabczak, A.: Electrochemical dressing of grinding wheels using alternating current. June 2000 International Journal of Abrasive Technology 3(6):16-20
  1. I recommend logically arranging the text, unambiguously declaring a new progress and changing the title of the publication (eg Electrochemical and X-ray examination of erosion products etc...).

Round 2

Reviewer 1 Report

The authors have improved the article according to the comments. The article can now be published.